# Microglia–Astrocyte Communication via C1q Contributes to Orofacial Neuropathic Pain Associated with Infraorbital Nerve Injury

**DOI:** 10.3390/ijms21186834

**Published:** 2020-09-17

**Authors:** Sayaka Asano, Yoshinori Hayashi, Koichi Iwata, Akiko Okada-Ogawa, Suzuro Hitomi, Ikuko Shibuta, Yoshiki Imamura, Masamichi Shinoda

**Affiliations:** 1Department of Oral Diagnostic Science, Nihon University School of Dentistry, 1-8-13 Kandasurugadai, Chiyoda-ku, Tokyo 101-8310, Japan; desa17001@g.nihon-u.ac.jp (S.A.); okada.akiko1@nihon-u.ac.jp (A.O.-O.); imamura.yoshiki@nihon-u.ac.jp (Y.I.); 2Department of Physiology, Nihon University School of Dentistry, 1-8-13 Kandasurugadai, Chiyoda-ku, Tokyo 101-8310, Japan; iwata.kouichi@nihon-u.ac.jp (K.I.); hitomi.suzuro@nihon-u.ac.jp (S.H.); shibuta.ikuko@nihon-u.ac.jp (I.S.); shinoda.masamichi@nihon-u.ac.jp (M.S.)

**Keywords:** microglia, astrocyte, C1q, infraorbital nerve injury, mechanical hypersensitivity

## Abstract

Trigeminal nerve injury causes a distinct time window of glial activation in the trigeminal spinal subnucleus caudalis (Vc), which are involved in the initiation and maintenance phases of orofacial neuropathic pain. Microglia-derived factors enable the activation of astrocytes. The complement component C1q, which promotes the activation of astrocytes, is known to be synthesized in microglia. However, it is unclear whether microglia–astrocyte communication via C1q is involved in orofacial neuropathic pain. Here, we analyzed microglia-astrocyte communication in a rat model with infraorbital nerve injury (IONI). The orofacial mechanical hypersensitivity induced by IONI was significantly attenuated by preemptive treatment with minocycline. Immunohistochemical analyses revealed that minocycline inhibited the increase in c-Fos immune-reactive (IR) cells and the fluorescence intensity of both Iba1 and glial fibrillary acidic protein (GFAP) in the Vc following IONI. Intracisternal administration of C1q caused orofacial mechanical hypersensitivity and an increase in the number of c-Fos-IR cells and fluorescence intensity of GFAP. C1q-induced orofacial mechanical hypersensitivity was completely abrogated by intracisternal administration of fluorocitrate. The present findings suggest that the enhancement in the excitability of Vc nociceptive neurons is produced by astrocytic activation via the signaling of C1q released from activated microglia in the Vc following IONI, resulting in persistent orofacial neuropathic pain.

## 1. Introduction

Trigeminal nerve injury due to tooth extraction, orofacial trauma, or dental implant displacement is known to cause persistent neuropathic pain in the orofacial region. Neuropathic pain is pathological in nature and persists for an extended period of time even after healing of the overt tissue damage [1]. Neuropathic pain in the orofacial area dramatically reduces quality of life by disturbing food intake, face-washing, and brushing of teeth. Thus, it is essential to clarify the mechanisms of trigeminal neuropathic pain to develop appropriate treatments.

Following trigeminal nerve injury, nociceptive neurons in the trigeminal spinal subnucleus caudalis (Vc) and upper cervical spinal cord (C1-C2) are excessively activated [2]. In accordance with excessive neuronal activity, microglial and astrocytic activation has been observed in the Vc and C1-C2 [3,4]. Microglial activation occurs for 1–3 days, whereas astrocytic activation takes 1–2 weeks after peripheral nerve injury [3,4]. Indeed, specific inhibitors of microglial and astrocytic activation markedly attenuate the initiation and maintenance phases of neuropathic pain, respectively [3,4]. Furthermore, pharmacological inhibition or pharmacogenetic ablation of microglia failed to ameliorate the maintenance phase of neuropathic pain [5,6,7]. Therefore, it is believed that microglia and astrocytes play an important role in the different phases of neuropathic pain [8]. It has been reported that microglia-derived interleukin (IL)-18 activates astrocytes in the spinal cord, resulting in the occurrence of tactile allodynia [9]. This evidence indicates that microglia–astrocyte communication is required for the development of neuropathic pain in the spinal cord. However, the involvement of microglia–astrocyte communication in orofacial neuropathic pain remains unknown.

C1q is a glycoprotein composed of 18 polypeptide chains consisting of three non-identical subunits, which bind to immune complexes containing immunoglobulins (Ig) G or IgM or to a variety of other activating substances, including C-reactive protein, retroviruses, and mitochondria [10]. After C1q binding, C1r and C1s are converted into proteolytic enzymes that are responsible for maintaining activation. Finally, the downstream complement protein forms a membrane attack complex [11]. A growing body of evidence indicates the importance of C1q in the central nervous system, and C1q is predominantly produced in microglia [12,13]. Recently, Liddelow et al. found that C1q induces A1 reactive astrocyte formation [14]. These reports suggest that C1q plays a role in microglia–astrocyte communication, which is important for understanding the sensitization mechanisms of Vc neurons.

The aim of this study was to clarify whether C1q activates astrocytes in the Vc, which in turn induces mechanical hypersensitivity in the orofacial region.

## 2. Results

### 2.1. Nocifensive Behavior and c-Fos Expression in the Vc Following IONI by Preemptive Treatment with Minocycline

The mechanical head withdrawal threshold (MHWT) was significantly reduced one day after infraorbital nerve injury (IONI) in saline-administrated rats, and this reduction in MHWT was observed throughout the experimental period. On the other hand, minocycline, an inhibitor of microglial activation, significantly attenuated the reduction of MHWT compared to saline administration (Figure 1A). To further clarify the effect of minocycline on neuronal activity, we analyzed c-Fos expression, a marker of neuronal activation, in the Vc. The number of c-Fos-IR cells was significantly increased seven days after IONI in saline-administrated rats (Figure 1B,C). The increase in the number of c-Fos-IR cells on day seven after IONI was significantly attenuated by minocycline administration (Figure 1B,C).

### 2.2. Effect of Minocycline Administration on Iba1 and GFAP Expression

The effects of minocycline on microglial activation and sequential activation of astrocytes in the Vc following IONI were further investigated. The expression of ionized calcium-binding adapter molecule 1 (Iba1) in the Vc of saline-administrated rats was markedly increased seven days after IONI (Figure 2A,B). Enlarged images showed that microglia in the Vc had activated morphology, such as hypertrophic and shortened processes. Preemptive treatment with minocycline significantly attenuated Iba1 expression in the Vc. Moreover, the morphology of microglia in the Vc displayed thin and branched processes, indicating that minocycline sufficiently inhibited microglial activation. The expression of glial fibrillary acidic protein (GFAP) in the Vc was also significantly increased in the Vc seven days after IONI. The increase in GFAP expression was significantly attenuated by minocycline administration (Figure 2C,D).

### 2.3. Effect of FC Administration on GFAP, Iba1, and c-Fos Expression

To evaluate whether deactivation of astrocytes causes suppression of Vc neuronal activity, the effect of intracisternal fluorocitrate (FC), a metabolic inhibitor of astrocytes, on c-Fos expression was assessed in IONI rats. We first addressed whether astrocytic activation caused by IONI was attenuated by FC administration. Increased GFAP fluorescence intensity was observed in the Vc seven days after IONI in PBS-administrated rats (Figure 3A,B), similar to that observed in Figure 2B. On day 7 after IONI, increased expression of GFAP in the Vc was significantly attenuated by FC administration (Figure 3A,B). On the other hand, increased Iba1 expression in the Vc following IONI was unchanged by FC administration (Figure 3C,D). Thus, we evaluated c-Fos expression in the Vc. The increase in the number of c-Fos-IR cells observed 7 days after IONI was significantly attenuated by FC administration (Figure 4A,B).

### 2.4. C1q Expression in the Vc

To investigate the role of C1q as a microglia-derived activation factor of astrocytes after IONI, we conducted immunohistochemical analyses of C1q. Robust C1q immunofluorescence was observed throughout the Vc 7 days after IONI, whereas it was not observed in the Vc of sham rats (Figure 5A,B). Double immunohistochemical staining revealed that C1q immunofluorescence exclusively colocalized with Iba1-IR but not GFAP-IR and NeuN-IR cells, indicating that microglia produce C1q in response to IONI (Figure 5C).

### 2.5. Effect of C1q Administration on Nocifensive Behavior, c-Fos, and GFAP-IR Cell Expression

In order to evaluate whether C1q activates astrocytes, which in turn facilitates neuronal activity in the Vc, recombinant C1q was intracisternally administered in naïve rats. A decline in MHWT was observed beginning 3 days after C1q administration and continued to decrease throughout the experimental period (Figure 6A). On the other hand, the MHWT remained unchanged following intracisternal administration of PBS (Figure 6A). Consistent with the behavioral data, the number of c-Fos-IR cells and GFAP expression were significantly increased by C1q administration (Figure 6B–E). Enlarged images showed hypertrophic astrocytic processes after C1q administration (Figure 6D). On the other hand, Iba1 expression and microglial morphology were not altered by C1q administration (Figure 6F,G).

### 2.6. Effect of FC Administration on C1q-Related Mechanical Hypersensitivity

To analyze the involvement of astrocytic activation in the Vc on C1q-induced mechanical hypersensitivity, we used FC to prevent astrocytic activation. Intracisternal administration of FC completely blocked the development of C1q-induced mechanical hypersensitivity in whisker pad skin (Figure 7A). We further conducted histological analyses on day 7 after C1q-administration in rats. The number of c-Fos-IR cells as well as the fluorescence intensity of GFAP were significantly reduced by FC administration (Figure 7B–E). On the other hand, intracisternal administration of both C1q and FC did not influence the morphology of microglia in the Vc (Figure 7F,G).

## 3. Discussion

The findings of this study can be summarized as follows: (1) The MHWT was significantly reduced after IONI. (2) The number of c-Fos-IR cells and the fluorescence intensity of Iba1, GFAP, and C1q were significantly increased in the Vc following IONI. (3) Minocycline administration significantly suppressed the number of c-Fos-IR cells as well as reduced the expression of Iba1 and GFAP. (4) C1q administration caused an increase in the expression of GFAP and promoted mechanical hypersensitivity in naïve rats. (5) FC administration suppressed C1q-induced mechanical hypersensitivity. These findings suggest that C1q is involved in the enhancement of Vc neuronal excitability via astrocytic activation, resulting in orofacial mechanical hypersensitivity associated with IONI.

A growing body of evidence indicates the importance of the complement cascade within the central nervous system, under both physiological conditions as well as pathophysiological conditions [12,15]. It is known that microglia-derived C1q is required for synapse elimination in dorsal lateral geniculate nucleus neurons during the developmental stage [13]. Similarly, microglia-mediated synapse elimination through C1q has also been observed in the spinal cord [16]. However, C1q has been shown to only influence spine density in the spinal cord of inflammatory pain model mice, but not in healthy mice [16]. Consequently, an intracisternal injection of C1q may not alter the spine density of Vc neurons. The role of the complement cascade in neuropathic pain has been clarified in the spinal cord. After peripheral nerve injury, mRNA for C1q, C3, and C4 in the complement cascade and C5a receptor were induced in spinal microglia [17]. C1q, C3, and C4 sequentially converted into terminal complement component C5, which was further converted into C5a and C5b [11]. Griffin et al. also found that C5-knockout mice, which lack the formation of C5a, abrogated the initiation but not the maintenance phase of neuropathic pain [17]. These findings suggest that C5a-mediated neuropathic pain is dependent on microglia, but not astrocytes. Other complement cascade proteins including C3a and the membrane attack complex, which is composed of C5b, C6, C7, C8, and C9, did not elicit neuropathic pain [17,18]. Considering the above evidence and our current data, C1q and C5a among the complement cascade proteins are causative factors that induce neuropathic pain. However, the evidence that C1q directly elicits neuropathic pain without sequential conversion into downstream molecules has not been reported elsewhere. In the current study, we did not observe any microglial activation in the Vc following intracisternal administration of C1q. This result implies that C1q is not converted into C5a in the Vc. Recent studies have suggested a direct effect of C1q during neuropathic pain. Frizzled 8, which controls the Wnt pathway, was found to be a novel binding site for C1q [19]. Moreover, the expression of Frizzed 8 was observed in astrocytes in the spinal cord [20]. Thus, C1q might directly bind to astrocytes and activate them, resulting in orofacial neuropathic pain. In the future, it will be important to assess the possible involvement of the C1q-Frizzled 8 pathway in Vc astrocytes in orofacial neuropathic pain.

It is unclear how microglia produce C1q after peripheral nerve injury. A possible candidate for C1q induction in microglia is IL-6, given that IL-6-stimulated macrophages induce C1q mRNA [21]. IL-6 mRNA is upregulated in the spinal cord after peripheral nerve injury [22,23]. Therefore, IL-6 might trigger C1q production in microglia after peripheral nerve injury

C1q transforms quiescent astrocytes into a reactive phenotype called A1 reactive astrocytes, which have been observed in several neurodegenerative diseases, including Alzheimer’s, Huntington’s, Parkinson’s, amyotrophic lateral sclerosis, and multiple sclerosis [14]. A characteristic feature of reactive astrocytes is the excessive expression of GFAP, resulting in morphological alterations such as hypertrophic processes [24]. In line with this alteration, intracisternal administration of C1q induced morphological alterations in Vc astrocytes. Reactive astrocytes influence synaptic transmission via multiple pathways. They secrete glutamate, adenosine triphosphate, and CCL2 [25,26]. CCL2 immediately potentiates both *N*-methyl-D-aspartate and α-3-hydroxy-5-methyl-4-isoxazole propionic acid receptor-mediated currents in the spinal cord [27]. Glutamatergic neurotransmission is also altered by glutamine synthetase in astrocytes [28]. In the trigeminal system, glutamine is released from astrocytes transported into primary afferent terminals of the trigeminal nerve, where glutamine is converted into glutamate, resulting in the enhancement of glutamate release [29]. Therefore, reactive astrocytes facilitate neuronal activity.

We observed a delayed onset of mechanical hypersensitivity following intracisternal administration of C1q. This might be attributed to the weak activation of astrocytes by C1q. Indeed, C1q induced 5 of 12 A1 astrocyte-specific genes, whereas tumor necrosis factor (TNF) α and IL-1α induced 10 and 12 genes, respectively [14]. Furthermore, the extent of gene induction by C1q is relatively low compared to that by TNFα and IL-1α [14]. Given the potency of C1q on astrocytic activation, it takes time for astrocytes to be fully activated by intracisternal administration of C1q. In the current study, we cannot conclude whether the activation state of astrocytes is changed after the stimulation with C1q; however, it is likely that C1q-induced mechanical hypersensitivity is mediated by reactive astrocytes-related signaling.

In conclusion, trigeminal nerve injury affects microglia and astrocyte activation, which causes strong activation of Vc nociceptive neurons. Activated microglia induce C1q after trigeminal nerve injury, and C1q released from microglia is thought to be involved in astrocytic activation in the Vc, which causes hyperactivity in Vc nociceptive neurons, resulting in orofacial neuropathic pain.

## 4. Materials and Methods

### 4.1. Animals

A total of 75 male Sprague-Dawley rats (6–8 weeks, 200–250 g) were purchased from Japan SLC (Hamamatsu, Japan). All rats were maintained on a 12 h light/dark cycle (light on at 7:00 a.m.) with food and water ad libitum at ambient temperature (23 ± 1 °C). All rats were handled at least 5 days before the behavioral testing to minimize stress. The experimental protocol was approved by the experimentation committee at Nihon University (protocol number: AP19DEN018-1), and the experiments were conducted according to the guidelines of the International Association of the Study of Pain [30].

### 4.2. Surgical Procedures

Rats were anesthetized with a mixture of butorphanol (2.5 mg/kg; Meiji Seika Pharma, Tokyo, Japan), midazolam (2.0 mg/kg; Sandoz, Tokyo, Japan), and medetomidine (0.15 mg/kg; Zenoaq, Fukushima, Japan). The IONI was performed according to a previously described method [31,32]. Briefly, a small incision was made at the left buccal mucosa, and the infraorbital nerve bundle was freed from the surrounding tissue. One-third of the nerve bundle was tightly ligated with 6-0 silk thread, and the incision was sutured. The rats that received the incision at the left buccal mucosa without nerve ligation were used as a sham-operated group. Finally, the rats were recovered with antisedan (Nippon Zenyaku Kogyo, Fukushima, Japan).

### 4.3. Intracisternal Administration

One day before IONI, a small hole was made in the occipital bone, and then the sterilized cannula (φ0.8 × 1.3 mm polyethylene tube) was carefully inserted into a hole along the occipital bone under anesthesia with a mixture of butorphanol, midazolam, and medetomidine described above. The tip of the tube was placed in a cisterna magna. Then, an osmotic pump (model 1002; Alzet, Cupertino, CA, USA) that was filled with saline, PBS, minocycline hydrochloride (Sigma-Aldrich, St. Louis, MO, USA), or FC (Sigma-Aldrich) was connected to the cannula. Minocycline hydrochloride and fluorocitrate were delivered into the intracisternal space at a rate of 4 nmol/h and 4 fmol/h, respectively. For the single administration, human recombinant complementary C1q (400 ng/5 μL, Abcam, Cambridge, MA, USA) or saline was injected through the cannula in naïve rats under deep anesthesia. Five microliters of FC (100 fmol) or PBS was conducted one day prior to the C1q injection. C1q, FC, or PBS were delivered once.

### 4.4. Behavioral Testing

Prior to the behavioral testing, all rats were trained to stay in the chamber, as described previously [31,32]. The MHWT was measured using von Frey filaments (4, 6, 8, 10, 15, 26, 30, 40, 50, and 60 g) applied to the whisker pad skin. Each von Frey filament was applied to the whisker pad skin 5 times. The lowest filament that elicited nociceptive responses such as head withdrawal and vocalization more than three times was deemed the MHWT. The MHWT in IONI rats and C1q-administrated rats was measured for seven consecutive days.

### 4.5. Immunohistochemistry

After behavioral testing, the rats were perfused transcardially with saline, followed by 4% paraformaldehyde in 0.1 M phosphate buffer under deep anesthesia with 5% isoflurane. The brainstem segments were further fixed by immersion in 4% paraformaldehyde overnight at 4 °C. The brainstem segments were then treated with 30% sucrose to prevent cryolesion. The Vc slices (30 μm thick) were made using a Cryostat (Sakura Finetek, Tokyo, Japan). The slices were treated with HistoVT One (Nacalai Tesque, Kyoto, Japan) at 70 °C for 20 min. Blocking was performed using PBS containing 0.3% Triton-X and 1% donkey serum for 1 h at room temperature. The sections were incubated with rabbit anti-Iba1 antibody (1:1000, 019-19741; FUJIFILM Wako, Osaka, Japan), rabbit anti-GFAP antibody (1:1000, Z0334, DAKO, Glostrup, Denmark), rabbit anti-c-Fos antibody (1:1000, ab190285, Abcam, Cambridge, MA, USA), rabbit anti-C1q (1:500, ab182451, Carlsbad, CA, USA), and goat anti-Iba1 antibody (1:1000; ab5076, Abcam), mouse anti-GFAP antibody (1:1000; MAB360, Merck Millipore, Burlington, MA, USA), or mouse anti-NeuN antibody (1:2000; MAB377B, Merck Millipore) for 2 days at 4 °C. After washing with PBS three times for 10 min, the slices were incubated with a secondary antibody conjugated with Alexa Fluor 488 or Alexa Fluor 594 (1:1000, Thermo Fisher Scientific, Waltham, MA, USA) for 2 h at room temperature. The slices were mounted in antifading medium (PermaFluor, Thermo Fisher Scientific, Waltham, MA, USA). Images were captured using a confocal laser-scanning microscope (LSM510; Carl Zeiss, Oberkochen, Germany). Images were analyzed using Image J software (http://rsbweb.nih.gov/ij/). For the counting of c-Fos-IR cells, the middle part of the Vc where the 2nd branch of the trigeminal nerve terminates was captured, and the total number of c-Fos-IR cells within the image was counted by Image J plugin cell count. Fluorescence Intensity of Iba1, GFAP, and C1q was measured as follows. Single plane 20× fluorescence images in the middle part of the Vc were collected at the same gain with an LSM510 equipped with LSM 5 Image Browser (Carl Zeiss, Oberkochen, Germany). The original 12-bit image data were converted into 8-bit using Image J. Region of interest was set in all regions of 20× fluorescence images (447.5 × 447.5 µm^2^ area). Five Vc slices from each rat were stained and analyzed. The raw integrated density of pixels in each image was measured by Image J, and then its average value was calculated. The average value of raw integrated density of pixels was normalized to 100 × 100 µm^2^ area.

### 4.6. Statistical Analyses

Data are represented as the mean ± SEM. Statistical analyses of the results were performed with two-way ANOVA with post hoc Bonferroni test (for behavioral analyses) and two-way ANOVA with post hoc Tukey’s test and unpaired Student’s *t*-test (for immunohistochemical analyses) using the GraphPad Prism 7 software package. A value of *p* < 0.05 was considered statistically significant.

## Figures and Tables

**Figure 1 ijms-21-06834-f001:**
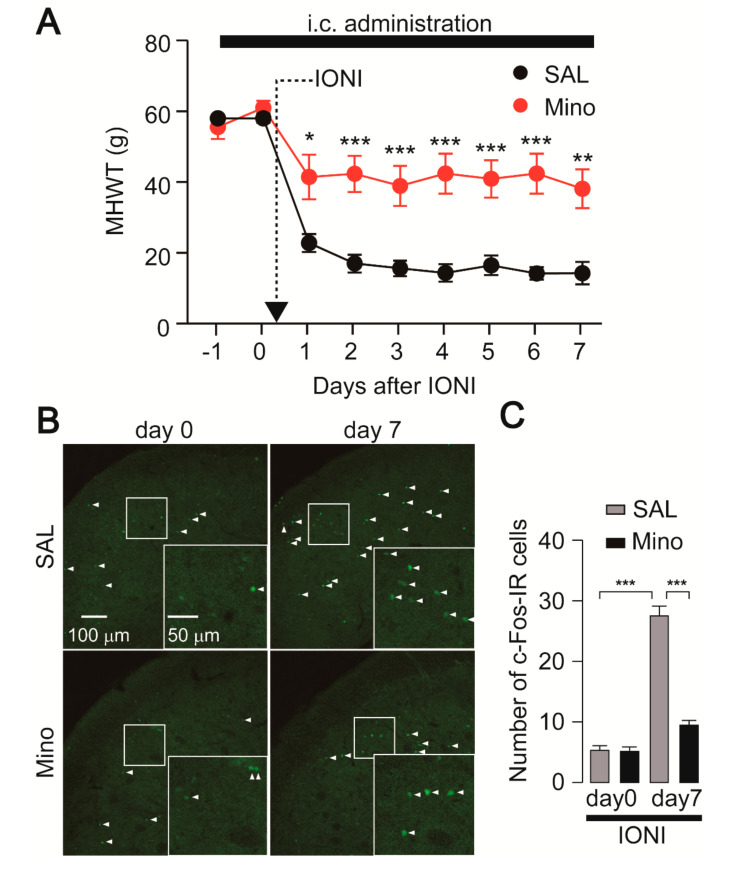
Minocycline inhibits the development of mechanical hypersensitivity in whisker pad skin and reduces c-Fos expression in the Vc following infraorbital nerve injury (IONI). (**A**) Time-course of mechanical head withdrawal threshold (MHWT) applying a von Frey filament to the whisker pad skin. SAL: n = 7, Mino: n = 10, two-way ANOVA post hoc Bonferroni test, * *p* < 0.05, ** *p* < 0.01, *** *p* < 0.001 (vs. SAL-treated group). SAL and Mino indicate saline and minocycline, respectively. Black bar indicates the period treated saline or minocycline. (**B**) Representative images of c-Fos immunofluorescence in the Vc 0 and 7 days after IONI. Insets indicate enlarged images of the region indicated in the open square. Arrowheads indicate c-Fos-IR cells. (**C**) The average number of c-Fos-IR cells in the Vc of saline or minocycline-treated rats 0 and 7 days after IONI. SAL (day 0): n = 5, Mino (day 0): n = 5, SAL (day 7): n = 7, Mino (day 7): n = 10, two-way ANOVA followed by Tukey’s multiple comparison test, *** *p* < 0.001. The data represent the means ± SEM.

**Figure 2 ijms-21-06834-f002:**
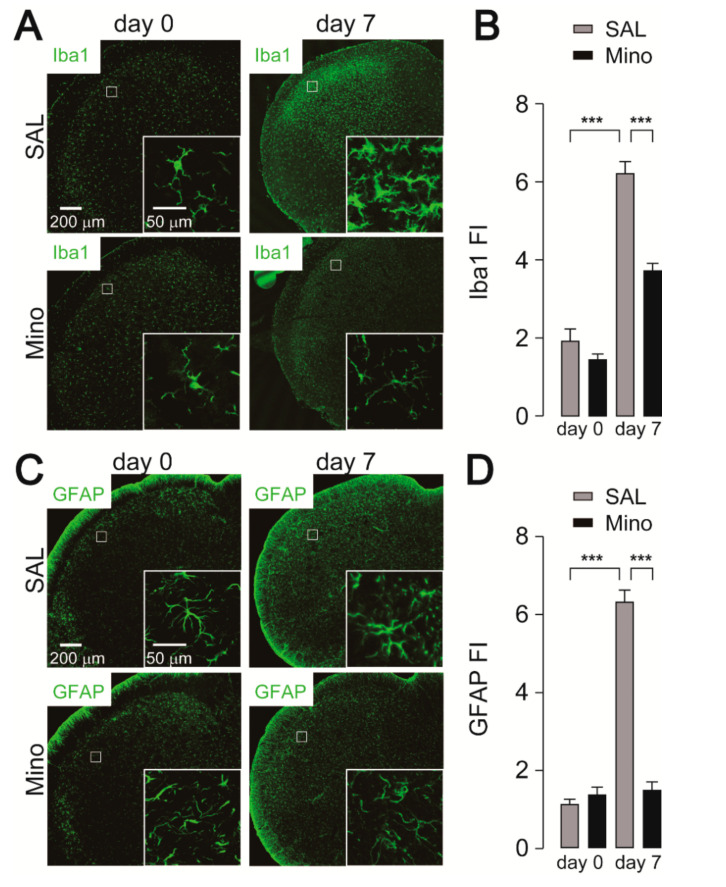
Inhibitory effects of minocycline on cellular activation of microglia and astrocytes in the Vc following IONI. (**A**) Representative images of Iba1 immunofluorescence in the Vc. Insets indicate enlarged images of the region indicated in the open square. (**B**) Fluorescence intensity (FI) of Iba1 in the Vc of saline or minocycline-treated rats 0 and 7 days after IONI. SAL (day 0): n = 5, Mino (day 0): n = 5, SAL (day 7): n = 7, Mino (day 7): n = 10, two-way ANOVA followed by Tukey’s multiple comparison test, *** *p* < 0.001. (**C**) Representative images of glial fibrillary acidic protein (GFAP) immunofluorescence in the Vc. Insets indicate enlarged images of the region indicated in the open square. (**D**) Fluorescence intensity (FI) of Iba1 in the Vc of saline or minocycline-treated rats 0 and 7 days after IONI. SAL (day 0): n = 5, Mino (day 0): n = 5, SAL (day 7): n = 7, Mino (day 7): n = 10, two-way ANOVA Tukey’s multiple comparison test, *** *p* < 0.001. The data represent the means ± SEM.

**Figure 3 ijms-21-06834-f003:**
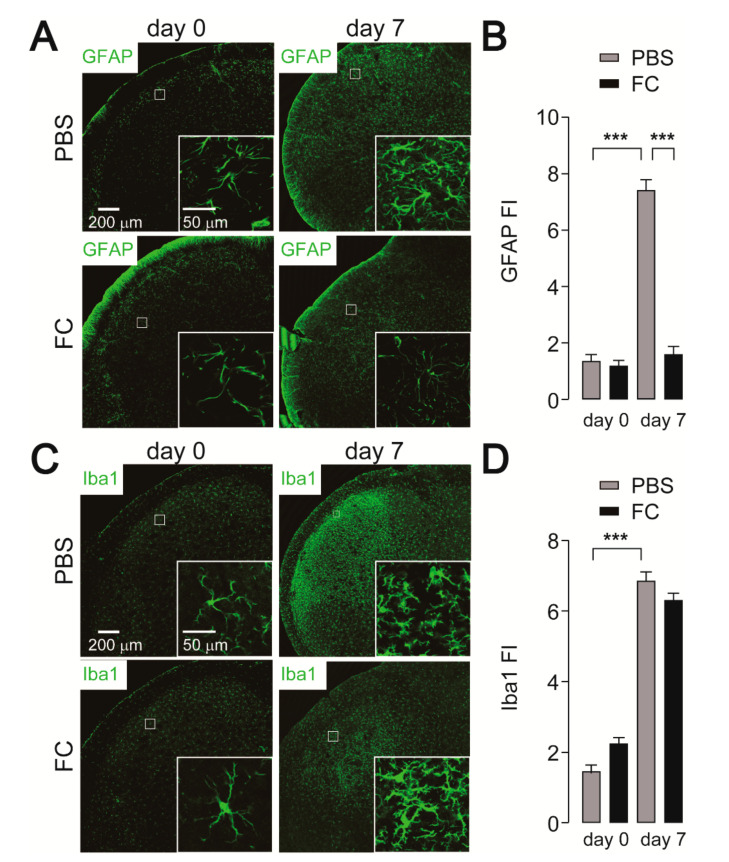
Effect of fluoroacetate on cellular activation of astrocytes and microglia in the Vc following IONI. (**A**) Representative images of GFAP immunofluorescence in the Vc. Insets indicate enlarged images of the region indicated in the open square. (**B**) GFAP fluorescence intensity (FI) in the Vc of saline or fluorocitrate (FC)-treated rats 0 and 7 days after IONI. PBS (day 0): n = 5, FC (day 0): n = 5, PBS (day 7): n = 5, FC (day 7): n = 5, two-way ANOVA followed by Tukey’s multiple comparison test, *** *p* < 0.001. (**C**) Representative images of GFAP immunofluorescence in the Vc. Insets indicate enlarged images of the region indicated in the open square. (**D**) Iba1 FI in the Vc of saline or FC-treated rats 0 and 7 days after IONI. PBS (day 0): n = 5, FC (day 0): n = 5, PBS (day 7): n = 5, FC (day 7): n = 5, two-way ANOVA followed by Tukey’s multiple comparison test, *** *p* < 0.001. The data represent the means ± SEM.

**Figure 4 ijms-21-06834-f004:**
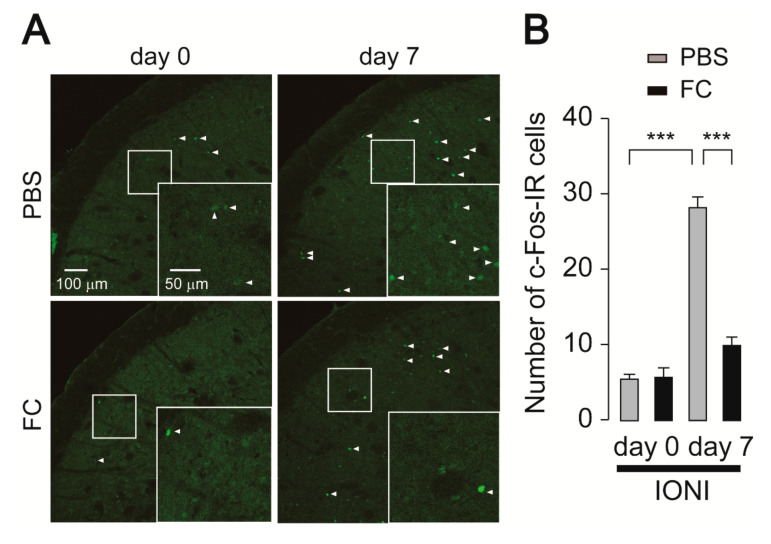
Fluorocitrate attenuates c-Fos expression in the Vc following IONI. (**A**) Representative images of c-Fos immunofluorescence in the Vc. Insets indicate enlarged images of the region indicated in the open square. Arrowheads indicate c-Fos-IR cells. (**B**) The average number of c-Fos-IR cells in the Vc of PBS or fluoroacetate (FC)-treated rats 0 and 7 days after IONI. PBS (day 0): n = 5, FC (day 0): n = 5, PBS (day 7): n = 5, FC (day 7): n = 5, two-way ANOVA followed by Tukey’s multiple comparison test, *** *p* < 0.001. The data represent the means ± SEM.

**Figure 5 ijms-21-06834-f005:**
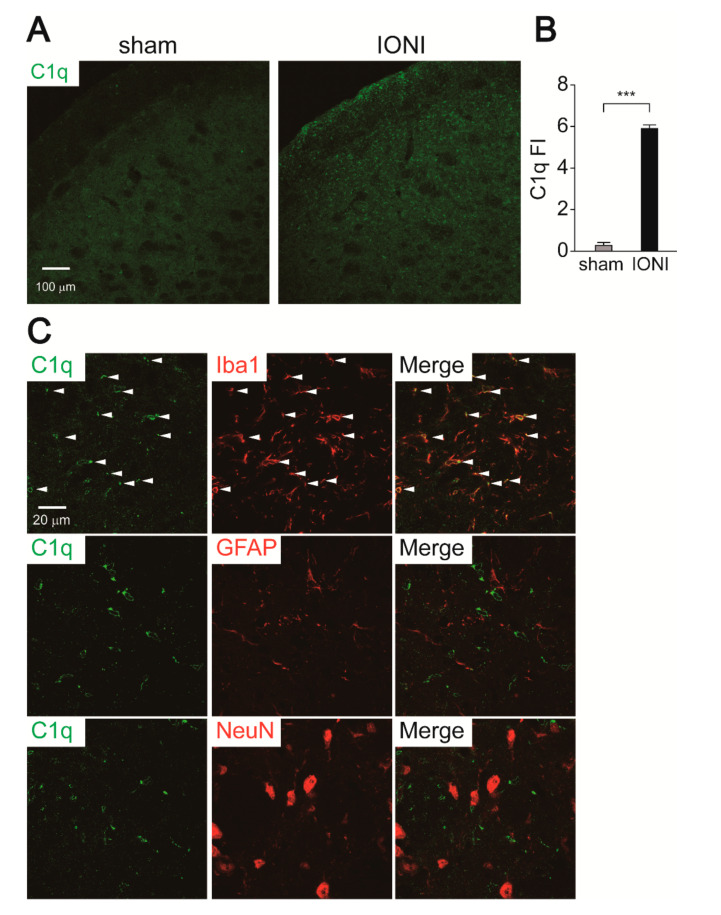
Induction of C1q in the Vc following IONI. (**A**) Representative images of C1q immunofluorescence in the Vc on day 7 of sham and IONI rats. (**B**) C1q fluorescence intensity (FI) in the Vc 7 days after IONI. Sham: n = 5, IONI: n = 5, unpaired Student’s *t*-test, *** *p* < 0.001. (**C**) The images show double staining of C1q and Iba1, GFAP, and NeuN in the Vc. Arrowheads indicate colocalization of C1q and Iba1 immunofluorescence.

**Figure 6 ijms-21-06834-f006:**
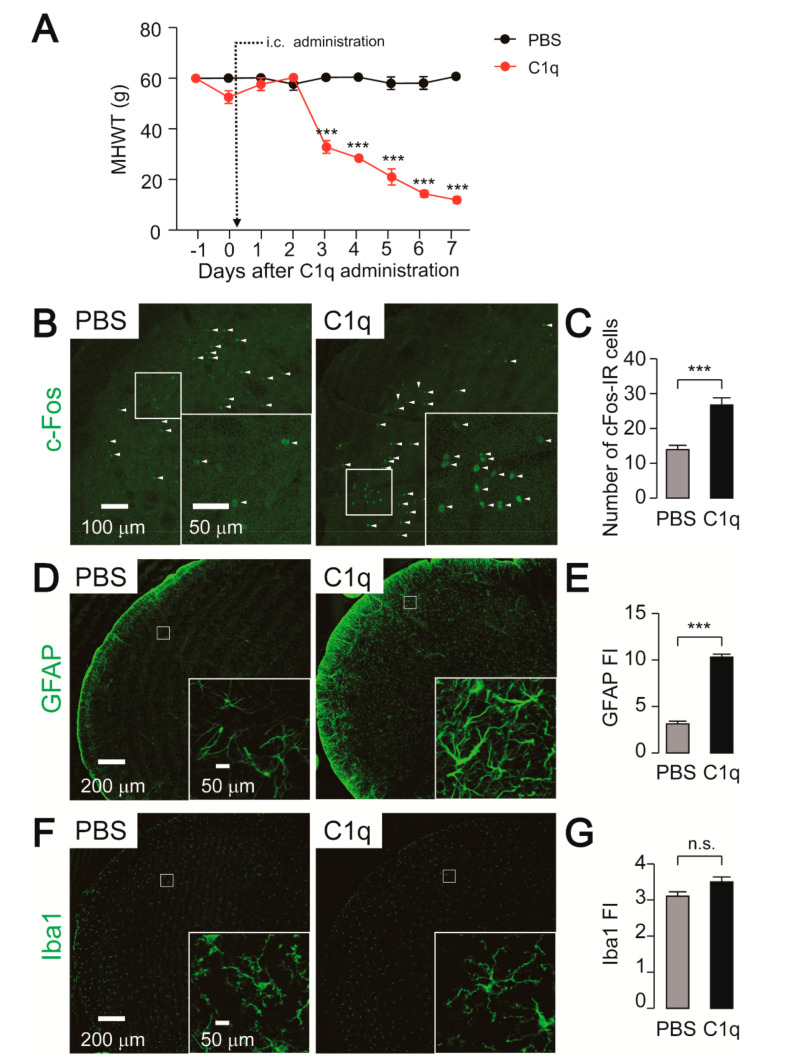
Intracisternal injection of C1q causes mechanical hypersensitivity and astrocytic activation. (**A**) Time-course of mechanical head withdrawal threshold (MHWT) in the whisker pad skin. PBS: n = 5, C1q: n = 5, two-way ANOVA post hoc Bonferroni test, *** *p* < 0.001 (vs. PBS-treated group). (**B**,**D**,**F**) Representative images of c-Fos (**B**), GFAP (**D**), and Iba1 (**F**) immunofluorescence in the Vc 7 days after PBS or C1q administration. Arrowheads indicate c-Fos-IR cells. Insets indicate enlarged images of the region indicated in the open square. (**C**,**E**,**G**) Averaged number of c-Fos-IR cells (**C**) and fluorescence intensity (FI) of GFAP (**E**) and Iba1 (**G**) in the Vc on day 7 of PBS or C1q-administrated rats. PBS: n = 5, C1q: n = 5, unpaired Student’s *t*-test, *** *p* < 0.001. n.s.: not significant. The data represent the means ± SEM.

**Figure 7 ijms-21-06834-f007:**
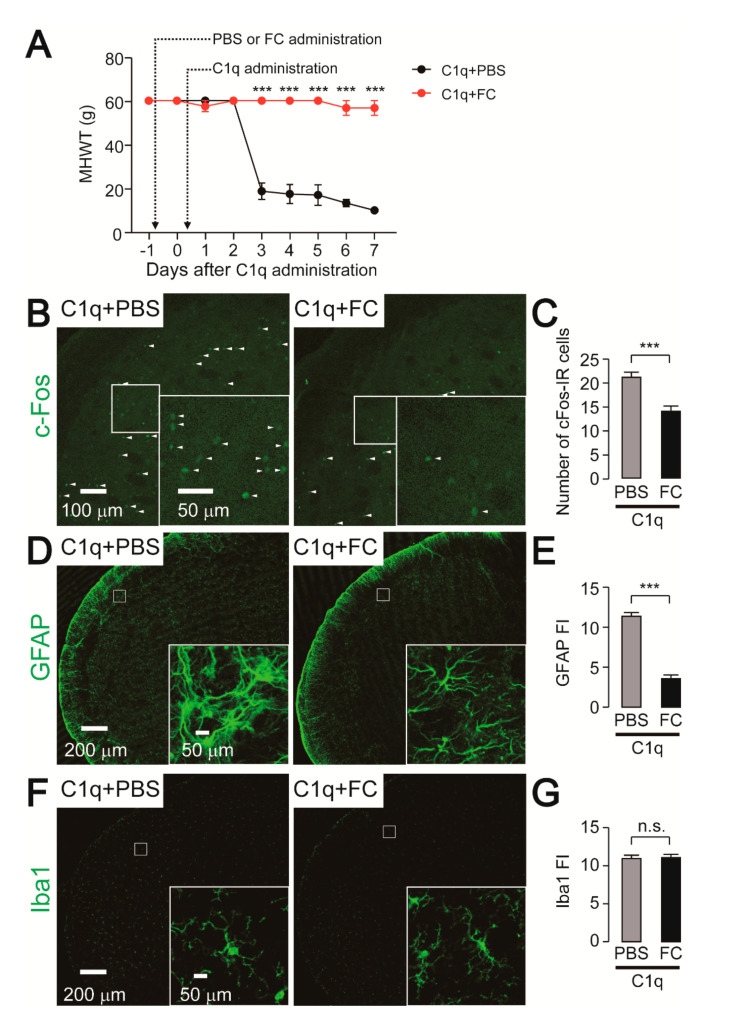
Inhibitory effects of FC on C1q-induced mechanical hypersensitivity and astrocytic activation. (**A**) Time-course of mechanical head withdrawal threshold (MHWT) in the whisker pad skin. C1q+PBS: n = 5, C1q+FC: n = 5, two way ANOVA post hoc Bonferroni test, *** *p* < 0.001 (vs. C1q+PBS-treated group). (**B**,**D**,**F**) Representative images of c-Fos (**B**), GFAP (**D**), and Iba1 (**F**) immunofluorescence in the Vc 7 days after C1q+PBS or C1q+FC administration. Arrowheads indicate c-Fos-IR neurons. Insets indicate enlarged images of the region indicated in the open square. (**C**,**E**,**G**) Averaged number of c-Fos-IR cells (**C**) and fluorescence intensity (FI) of GFAP (**E**) and Iba1 (**G)** in the Vc on day 7 of C1q+PBS or C1q+FC-administrated rats. C1q+PBS: n = 5, C1q+FC: n = 5, unpaired Student’s *t*-test, *** *p* < 0.001. n.s.: not significant. The data represent the means ± SEM.

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
