# Peer review of "Microglia–Astrocyte Communication via C1q Contributes to Orofacial Neuropathic Pain Associated with Infraorbital Nerve Injury"

_ijms, 2020, doi:10.3390/ijms21186834_

Round 1

Reviewer 1 Report

In this study, Asano et al. showed the increased C1q in the trigeminal spinal nerve (Vc) is tightly related to the orofacial neuropathic pain following nerve injury. According to the study, the infraorbital nerve injury (IONI) promoted the microglia to synthesize C1q, and this complement component evoked astrocyte activation, leading to the hyperactivity of the cells shown by cFos IR.  The authors also demonstrated the prior treatment of fluorocitrate could attenuate the increase of C1q as well as mechanical allodynia symptoms after the nerve injury. Overall, the study was appropriately designed and appears to be well-executed. The findings of the authors are rigorous and interesting, and the manuscript is well-written. I have only a few remarks.

  1. In figure 5, the authors demonstrated the increased C1q in the Vc after the IONI. These data are one of the core founding but seems like the detailed explanation and analysis are missing. It would be nicer if the authors could quantify the expression of C1q in the Vc following IONI, and perform statistical analysis to show the numerical difference between groups.
  2. In the same figure – what was the number of the samples used in these experiments? The authors did not provide detailed information in the main text or the figure legend.
  3. I was a bit curious regarding the demonstration of the fluorescence intensity (FI) of Iba1 or GFAP. How was the FI calculated? Was that an arbitrary unit? The authors described that they used ImageJ to measure the FI in the 100 * 100 micrometer square regions. Was the region sampled from slices from multiple rats or did they use multiple slices from 1~2 rats?
  4. I also wondered regarding the difference of baseline values, i.e. day 0 control groups in figure 2, 3, and the PBS control group in figure 6. For example, the baseline value of GFAP is around 1~1.5 in figure 2D and 3B but seems like >3 in figure 6E which is 2~3x higher than the data of figure 2 and 3.

Reviewer 2 Report

The authors provide evidence for the role of microglial-derived factor (C1q) in orofacial pain. The results support the role of C1q in microglia-astrocyte communication and orofacial mechanical hypersensitivity associated with infraorbital nerve injury. The presented findings suggest that the hyper-excitability of subnucleus caudalis neurons is produced by astrocytic activation via the signaling of C1q released from activated microglia in the subnucleaus caudalis of rats exposed to infraorbital nerve injury. This is a very nice, concise and thorough study providing evidence for the importance of the complement cascade within the central nervous system and trigeminal neuropathic pain. The results are presented very clearly. The following are my minor comments:

Some minor comments:

  1. Could you please correct a sentence line 44 "Microglial activation occurs during 1 from 3 days, (...)"? Suggestion "Microglial activation occurs duing 1-3 days, (...)"
  2. Could you please correct a typing error line 96 “hypertorophic”? I think it should read hypertrophic.
  3. Could you please clarify how frequently was C1q delivered?

In section 4.3. Intracisteranl administration, Line 286 it is stated: “Five uL of FC (100fmol) or PBS was injected prior to the C1q injection.” In the following section 4.4 Behavioral Testing: lines 292 and 293, it is stated: “the MHWT in IONI rats and C1q administered rats was measured for seven consecutive days.” It is not clear if the C1q was delivered once or daily for seven days. Could you please clarify?
